# Why Were [GADV]-amino Acids and GNC Codons Selected and How Was GNC Primeval Genetic Code Established?

**DOI:** 10.3390/genes14020375

**Published:** 2023-01-31

**Authors:** Kenji Ikehara

**Affiliations:** 1G&L Kyosei Institute, The Keihanna Academy of Science and Culture (KASC), Keihanna Interaction Plaza, Lab. Wing 3F, 1-7 Hikaridai, Seika-cho, Souraku, Kyoto 619-0237, Japan; ikehara@cc.nara-wu.ac.jp; 2International Institute for Advanced Studies, Kizugawadai 9-3, Kizugawa, Kyoto 619-0225, Japan

**Keywords:** origin of the genetic code, GNC code frozen-accident theory, coevolution theory, adaptive theory, origin of tRNA, GADV hypothesis, origin of life, core life system

## Abstract

Correspondence relations between codons and amino acids are determined by genetic code. Therefore, genetic code holds a key of the life system composed of genes and protein. According to the GNC-SNS primitive genetic code hypothesis, which I have proposed, it is assumed that the genetic code originated from GNC code. In this article, first, it is discussed from a standpoint of primeval protein synthesis, why four [GADV]-amino acids were selected and used in the first GNC code. Next, it is explained from another standpoint of the most primitive anticodon-stem loop tRNAs (AntiC-SL tRNAs), how four GNCs were selected for the first codons. Furthermore, in the last section of this article, I will explain my idea of how the correspondence relations between four [GADV]-amino acids and four GNC codons were established. Namely, the origin and evolution of the genetic code was discussed comprehensively from several aspects of [GADV]-proteins, [GADV]-amino acids, GNC codons, and anticodon stem-loop tRNAs (AntiC-SL tRNAs), which relate each other to the origin of the genetic code, as integrating GNC code frozen-accident theory, coevolution theory, and adaptive theory on the origin of the genetic code.

## 1. Introduction

The genetic code connecting gene with protein is one of the six members (gene, tRNA, genetic code, protein, metabolism, and cell structure) composing the fundamental life system, including the core life system (gene, tRNA, genetic code, and protein) [1]. Revealing the origin of the genetic code, one of four members of the core life system might lead to solving the mystery of the origin of life (Figure 1) [1]. Therefore, it is quite important to clarify how amino acids and codons were selected among messy organic compounds on primitive Earth and how the first genetic code was established.

There is no established theory on the origin of the genetic code, although many studies were carried out by many researchers and several hypotheses have been proposed such as GCU theory [2], GC theory [3], four column theory [4], RNY theory [5,6], and so on, thus far. Those studies were mainly carried out through analyses of codon usage [2,5,6], the early metabolism [3], and amino acid synthesis with prebiotic means [4]. However, a correct answer to the origin of the genetic code could not be obtained as long as the studies are carried out without taking protein formation into consideration because genetic code is intimately related to formation of proteins.

On the contrary, I proposed the GNC-SNS primitive genetic code hypothesis [7], suggesting that the universal genetic code originated from GNC primeval genetic code via SNS primitive genetic code [8]. SNS stands for G and C. The hypothesis was triggered by the GC-NSF(a) hypothesis for entirely new gene formation, which was obtained by exploring conditions for water-soluble globular proteins about 20 years ago [9]. GC-NSF(a) means nonstop frame on antisense strand of GC-rich gene. Therefore, the hypothesis was proposed as considering the formation process of water-soluble globular proteins. 

In this article, I further present grounds supporting the idea advocating that the universal (standard) genetic code originated from GNC code. For this purpose, it is important to understand that there are two aspects of the origin of the genetic code. Those are the following two. 

From what code did the genetic code originate?How were amino acids and codons selected for the first genetic code and how was the first genetic code established?

Then, I explain the former aspect of the origin of the genetic code. 

## 2. GNC-SNS Primitive Genetic Code Hypothesis

Only main points are described here as the details are described in a paper published by Ikehara et al. [7].

### 2.1. GC-NSF(a) Hypothesis for Formation of Entirely New Genes

The study on the GADV hypothesis started from the origin of modern genes independently of the origins of the genetic code and life. Consequently, the GC-NSF(a) hypothesis was obtained, which suggests that entirely new genes are created from nonstop frames on GC-NSF(a)s. Note that those studies were carried out with amino acid composition and six properties of the respective amino acids, or hydropathy, α-helix, β-sheet, turn/coil formabilites, and acidic and basic amino acid compositions, all of which are related to water-soluble globular protein formation.

### 2.2. SNS Primitive Genetic Code Hypothesis

Next, base compositions at three codon positions were analyzed to reveal the reason why entirely new genes could be generated from an essentially random codon sequence encoded by GC-NSF(a). From the results, it was found that G and C are naturally used at high frequencies at the first and the third codon positions in the GC-rich region (Figure 2A,B,D), but curiously, four bases, G, C, A, and T, are used at similar frequencies at the second base position of codon of GC-rich genes (Figure 2C) and that, in other words, codon sequences similar to (SNS)_n_ are written on antisense strands of GC rich genes (Figure 2) [9].

It was further analyzed, whether or not imaginary proteins encoded by SNS code, which were generated on a computer, can satisfy the six conditions (hydrophobicity/hydrophilicity or hydropathy, α-helix, β-sheet, turn/coil formabilities, and acidic and basic amino acid compositions) for water-soluble globular protein formation. From the results, it was found that even the imaginary polypeptides using only ten amino acids encoded by SNS code could be folded into water-soluble globular structures at a high probability. Based on the results, we proposed the SNS primitive genetic code hypothesis [8]. 

### 2.3. GNC Primeval Genetic Code Hypothesis

Furthermore, I explored whether or not more primitive imaginary proteins, which use only four amino acids extracted from columns and rows of the universal genetic code, can satisfy the four conditions (hydropathy, α-helix, β-sheet, and turn/coil formabilites), in which two other conditions (acidic and basic amino acid compositions) are excluded because of less importance than others. It became clear that essentially only one code composed of four GNC codons and four [GADV]-amino acids could satisfy the four conditions, except GNG code, in which Glu is used instead of Asp and encodes four [GAVE]-amino acids. Then, I presented the GNC-SNS primitive genetic code hypothesis (Figure 3) [7].

However, it must be understood that the GNC primeval genetic code hypothesis was obtained by analyses within twenty amino acids encoded by the universal genetic code. Therefore, the reason why four [GADV]-amino acids were selected and used in the GNC code composing four GNC codons, was unknown. Then, I decided to make the reason known, as further described in the next section.

In this article, the establishment process of the GNC primeval genetic code is discussed with a focus on the latter aspect, “How were amino acids and codons selected for the first genetic code and how was the first genetic code established?”, as summarizing the results obtained so far.

## 3. Why Were [GADV]-amino Acids and GNC Codons Selected?

It is expected that [GADV]-amino acids, which satisfy the four conditions for protein formation, were selected and used in the GNC genetic code because immature [GADV]-proteins could be produced by direct random joining of [GADV]-amino acids, even when both genes and genetic code had not been formed. Contrary to that, the other five members (cell structure, metabolism, tRNA, genetic code, and gene) previously require the existence of [GADV]-amino acids and/or immature [GADV]-proteins [1]. 

### 3.1. How Were Four [GADV]-amino Acids Selected?

It is well known that various amino acids, including simple β-amino acids and γ-amino acids, can be produced other than natural amino acids under experimental conditions carried out by Miller (Table 1) [10]. It is also confirmed by experiments carried out by Cleaves et al., in which oxidative gas was used instead of the reductive gas used in Miller’s experiments, that various amino acids were similarly synthesized [11]. Therefore, it becomes a problem how [GADV]-amino acids could be selected out from the messy amino acids, which accumulated on primitive Earth at large amounts. I will give an answer to the problem below.

#### 3.1.1. Why Are Only α-amino Acids Used in Proteins?

Only α-amino acids are used in the universal genetic code. The reason is because the use of only α-amino acids restricts free rotation to two bonds neighboring a carbon atom, which are sandwiched in between two peptide bonds having considerable double-bond character which prevents free rotation. On the contrary, use of β-amino acids and γ-amino acids make it difficult to form regular structures as α-helix and β-sheet structures because of additional free rotations of chemical bonds between α-carbon and β-carbon atoms and between β-carbon and γ-carbon atoms. Therefore, only α-amino acids must be used in protein synthesis. Inversely stating this, proto-cells using only α-amino acids could have a high productivity and left many descendants.

#### 3.1.2. Why Are Twenty Natural Amino Acids Used in Proteins?

It is also well known that various types of simple α-amino acids as 2-aminobutylic acid (2-ABA: α-amino-n-butylate) and norvaline are also synthesized in Miller’s experiments (Table 1) [10]. However, it has previously been unknown why those nonnatural α-amino acids are not used in proteins. The reason is explained below.

Amino acids having more than two methyl groups or a bulky side chain like a benzene ring of phenylalanine on β-carbon atom should have a large propensity for β-sheet formation because of a possible steric hindrance preventing α-helix formation. On the contrary, amino acids having two or three hydrogen atoms on a β-carbon atom should have a large propensity for α-helix formation (Table 2). Therefore, use of 2-ABA or norvaline without any bulky side chain on a β-carbon atom instead of valine having two methyl groups on a β-carbon atom causes an unfavorable excess of α-helix formability and, in parallel, insufficient β-sheet formability of proteins synthesized under an imaginable GNC code encoding three [GAD]-amino acids+2-ABA or norvaline. Thus, four [GADV]-amino acids were selected to effectively produce water-soluble globular proteins having appropriate hydropathy and three secondary structure formabilities as α-helix, β-sheet, and turn/coil formabilities (Table 2) [7]. 

#### 3.1.3. Why Are Only L-Amino Acids Used in Proteins?

Not only L-amino acids, but also D-amino acids, should be synthesized on primitive Earth without any asymmetrical field. Nevertheless, only L-amino acids are used in natural proteins. In other words, twenty amino acids are homochiral. In this case too, amino acids, which are used in proteins, must be homochiral. Secondary structures or regular structures are unsuccessfully formed if both L-amino acids and D-amino acids are used in protein synthesis, because propensity of secondary structure formation of D-amino acids is opposed to that of L-amino acids. On the other hand, Gly without asymmetric carbon atom is used for inhibiting secondary structure formation and promoting turn/coil formation of [GADV]-proteins (Table 2).

The homochirality of three [ADV]-amino acids could be attained by differential crystallization, which accidentally occurred between racemic amino acid mixtures during the drying process of the [GADV]-amino acid aqueous solution under sunlight on primitive Earth [12,13]. 

**Table 2 genes-14-00375-t002:** Propensities for hydropathy, secondary structure formations of [GADV]-amino acids. Four [GADV]-amino acids with a simple chemical structure could be easily synthesized with prebiotic means (Table 1). Nevertheless, a hydrophilic Asp and a hydrophobic Val are contained in the four amino acids. In addition, α-helix forming Ala, β-sheet forming Val, and turn/coil forming Gly are also contained in the four amino acids. Numerals indicate the respective propensities quoted from Stryer’s textbook “*Biochemistry*” [14]. The splendid properties of [GADV]-amino acids make it possible to satisfy the four conditions for water-soluble globular protein formation. That is the reason why [GADV]-amino acid composition is one of protein 0th-order structure.

	Gly [G]	Ala [A]	Asp [D]	Val [V]
Hydropathy	1	1.6	−9.2	2.6
α-Helix	0.56	1.29	1.04	0.91
β-Sheet	0.92	0.9	0.72	1.49
Turn/coil	1.64	0.78	1.41	0.47

#### 3.1.4. Why Are Hydrophobic Val and Hydrophilic Asp Encoded in the GNC Code? 

The existence of both hydrophobic Val, which forms the hydrophobic core structure in a protein, and hydrophilic Asp, which is favorable to locate on the surface of a protein, are indispensable to form a stable globular structure in water. Thus, usage of three L-α-[ADV]-amino acids plus Gly is necessary to form water-soluble globular proteins. Furthermore, it was confirmed that four [GADV]-amino acids are the simplest combination among twenty natural amino acids, with which water-soluble globular protein can be effectively formed [15]. Therefore, cell structures using four [GADV]-amino acids were selected as the results of repeated “trial and error” or of natural selection among the proteins using various types of amino acids, which accumulated on the primitive Earth in large amounts. 

Naturally, proteins before formation of genetic information must be produced through random processes. Therefore, the most primitive proteins must be produced by the direct random joining of [GADV]-amino acids under one of protein 0th-order structures (Figure 4) [16]. That is the only way for the production of immature but meaningful proteins leading to the emergence of life. This was confirmed by analysis, in which points were given when imaginary [GADV]-proteins satisfied the four conditions for water-soluble globular protein synthesis [1,7,15].

It is well known that [GADV]-amino acids are easily synthesized by Miller-type experiments (Table 1) [11,12] and are detected at large amounts from Murchison meteorite [4,17]. Therefore, the GADV hypothesis is consistent with the results obtained by Miller-type experiments and the results of chemical analyses of the meteorites, although the hypothesis is not founded on sufficient experimental results at this point in time. Further, it is also confirmed that [GADV]-amino acid composition is the simplest combination among twenty amino acids, which satisfies the four conditions for water-soluble globular protein formation [15]. 

Thus, steps to the emergence of life began upon the formation of immature [GADV]-proteins. Formations of GNC primeval genetic code and the first (GNC)_n_ gene succeeded the immature [GADV]-protein formation as aiming at more efficiently producing [GADV]-proteins with higher functionality step by step. The first genetic code and gene became established as the results of repeated “trial and error” or through selection of [GADV]-microspheres, which could grow and proliferate faster than before through acquisition of immature but more and more efficient [GADV]-proteins [1]. 

The GNC primeval genetic code hypothesis, suggesting from which the universal genetic code originated from the GNC code, is consistent with the idea, which was described in Watson’s textbook [5], and in the results obtained by Shepherd [6]. Therefore, I am convinced that the universal genetic code originated from the GNC code.

However, the GNC code described above is only the result that was obtained by investigation in a frame of the universal genetic code, which was triggered by a study on where and how entirely new genes are generated in extant organisms. Therefore, the reason is not evident why GNC codons were used in the first genetic code.

### 3.2. How Were Four GNC Codons Selected for the First Genetic Code?

Next, consider the reason how four GNC codons were selected and used in the first genetic code as suggested by the GNC-SNS primitive genetic code hypothesis [7]. 

#### 3.2.1. Grounds Showing That GNC Codons Were Used in the First Genetic Code

Of course, [GADV]-proteins produced by the direct random joining of [GADV]-amino acids is essentially the same as polypeptide chains synthesized with random (GNC)_n_ codon sequences, because both the proteins and polypeptides have a random [GADV]-amino acid sequence.

On the other hand, it is impossible to consider the formation process of the first genetic code connecting codons with amino acids independently of tRNAs, which actually mediate between codons and amino acids. It is then explained how the first tRNA was generated and what anticodons were used in the first tRNA. Naturally, the first tRNA must also be formed through random processes in the absence of genes. In this regard, I have proposed anticodon stem-loop (AntiC-SL) tRNA hypothesis on origin of tRNA [18], suggesting that modern tRNAs originated from [GADV]-AntiC-SL tRNAs, which were formed as the smallest but sufficiently stable hairpin loop RNAs composed of seventeen nucleotides through repeated random joining of nucleotides and degradation of oligonucleotides. It is obvious that the AntiC-SL tRNAs are sufficiently stable against RNase activity of immature [GADV]-proteins, because AntiC-SLs of modern *Escherichia coli* L-form tRNAs carrying a GNC anticodon that are not chemically modified except Asp-tRNA modified with a small methl group (/) and queosine (Q) (Table 3) [19].

#### 3.2.2. Strong Binding of a Triplet, GNC, with the Complementary Triplet, GNC 

Furthermore, it is also known that complementary triplet pairs, ^5′^GNC^3′^/^3′^CNG^5′^, are more stable than any other complementary triplet pairs except Ser (Table 4), as described in the paper published by Taghavi et al. (2017) [20], that *a series of codons of the form GNC (where N means ‘anything’) should have the special property of partitioning naturally at the codon boundary C to G when under tension* (Table 4). Their observations are supported by the fact that a GNC anticodon carried by an AntiC-SL binds with the complementary codon in mRNA during translation (Figure 5). 

## 4. How Were the Correspondence Relations between GNC Codons and [GADV]-amino Acids Established?

### 4.1. Direct Complex Formation between GNC Anticodons/Codons and [GADV]-amino Acids Is Impossible

Here, it is considered whether or not complexes can be formed between GNC anticodons and [GADV]-amino acids, and whether or not the complexes can be used for [GADV]-protein synthesis, if the complexes could be formed.

(1) Triplet GNC codons could not be directly bound with the respective [GADV]-amino acids as suggested by the stereochemical theory, which was proposed by Shimizu [21], because of the sizes of triplet GNC nucleotides. Even when the triplets were folded into a compact tertiary structure, they were too large to bind with small side chains as H-atom of Gly and methyl group of Ala. This is supported from the results obtained by Yarus [22] that stereospecific interaction between a triplet codon and its cognate amino acid with a small side chain as Gly, Ala, and so on could not be detected in complexes of RNA with proteins as ribosomes and riboswitches [22]. 

(2) Amino group and/or carboxyl group of [GADV]-amino acids could not expose outside from the complexes between anticodons and the corresponding amino acids, even if triplet nucleotides could be bound with [GADV]-amino acids, because those groups must be used for stable complex formation with triplet nucleotides. This means that the amino group and/or carboxyl group of [GADV]-amino acids cannot participate in [GADV]-protein synthesis. 

(3) In addition to that, the binding mode of four complexes of GNC codons with [GADV]-amino acids must be the same, because otherwise it becomes impossible to form a peptide bond between two neighboring [GADV]-amino acids in the complexes. This also indicates that it would be impossible to synthesize immature [GADV]-proteins having various amino acid sequences by using the complexes of GNC codons/anticodons with [GADV]-amino acids. 

(4) Furthermore, if the direct complexes between GNC codons and [GADV]-amino acids were used for [GADV]-protein synthesis, the direct stereochemical recognition system using complexes between GNC codons and [GADV]-amino acids for protein synthesis must, one day, be transferred to the indirect protein synthetic system using tRNA. However, it would be impossible to transfer the direct system to the indirect system with tRNA [1]. 

As explained so far, it is considered at this point in time that four [GADV]-amino acids (Section 3.1) and four GNC codons (Section 3.2) were selected for establishment of the GNC code. However, the reason why the correspondence relations between GNC codons and [GADV]-amino acids were determined is still entirely unknown. Inversely stating this, if it is considered from the standpoint of stereochemical theory, it must be shown that a historical trajectory from the direct joining to indirect joining of [GADV]-amino acids under the first genetic code must be explained. 

Therefore, the above considerations clearly indicate that the GNC primeval genetic code could not be established as the stereochemical theory assumes. In order to overcome the difficulties, it is important to understand how the correspondence relations between four GNC codons and four [GADV]-amino acids were determined during formation of the most primitive but specific AntiC-SL tRNAs. Then, I would like to give an answer to the second problem of how the GNC primeval genetic code was established, which is described in the introduction.

### 4.2. GNC Code Frozen-Accident Theory on the Origin of the Genetic Code

Then, how was the first GNC code established? During the above considerations, the GNC code freeze-accident theory was conceived, suggesting that only the correspondence relations between GNC codons and [GADV]-amino acids were accidentally determined and frozen after the correspondence relations were established [1]. The establishment process of the primeval genetic code could be consistently explained by incorporating the GNC code freeze-accident theory [1]. I firmly believe now that the GNC primeval genetic code was established as assumed by the GNC code frozen-accident theory [1], because it is considered that there is no other way. However, a problem still remains unsolved of how corresponding relations between GUC-Val, GCC-Ala, GAC-Asp, and GGC-Gly were formed, because there exists 24 combinations (4! = 24) between GNC codons and [GADV]-amino acids, even under the assumption that use of two pairs between Gly-Ala and Asp-Val were inevitable. The assumption is based on formation of entirely new [GADV]-protein synthesis, which should be carried out under the GNC code and (GNC)_n_ genes (Figure 5). Therefore, the last problem about the origin of the genetic code is how the four combinations, GUC-Val, GCC-Ala, GAC-Asp, and GGC-Gly, were selected and used in the GNC code, which evolved to the universal genetic code. I consider the problem as follows. 

### 4.3. How Was the First GNC Code Established?

In order to completely solve the problem of how the first GNC code was established, it is necessary to clarify how the correspondence relations of four GNC codons and four [GADV]-amino acids were selected among 24 ways of combinations. Here, we must pay enough attention to the fact that selection of combinations is not always completely random, because any mature [GADV]-proteins having a rigid and compact structure must always be formed from an immature [GADV]-protein with a flexible and swollen structure (Figure 6). Therefore, it means that two amino acid pairs, Val-Asp and Ala-Gly, must use either one of two complementary pairs, GUC-GAC and GCC and GGC. The combinations are restricted in eight ways. Therefore, the combinations, Val-GUC, Asp-GAC, Ala-GCC, and Gly-GGC, which were accidentally selected out among the eight ways and frozen, is the GNC code (Figure 3). 

## 5. Discussion

Many researchers have tried to make clear the origin of the genetic code from various viewpoints as described below.

From what code did the genetic code originate?How was the correspondence relations between codons and amino acids determined?

However, no researcher could give any clear answers to the two questions above thus far. On the contrary, I have advocated for the GNC primeval genetic code hypothesis as an answer to the first question that the genetic code originated from the GNC code (Figure 3) [7]. 

In order to solve the second question above, at the outset, we must answer to two questions described below. 

How were the four [GADV]-amino acids selected among messy amino acids which accumulated on primitive Earth?Why were the four GNC codons selected through random processes?

I could also give reasonable answers to the two questions (Section 3) as described in this article. In addition, both amino acids, which are encoded by GNC code and amino acids, which were synthesized in the earliest phase of amino acid metabolism, were [GADV]-amino acids [1]. Therefore, the idea that amino acids were used in the first genetic code, GNC, is consistent with the coevolution theory [23,24].

Hydrophobic property of an amino acid does not change largely by base substitution at the first base position of codon.Hydrophobic property of an amino acid changes largely by base substitution at the second base position of codon.In many cases, the same amino acid is used in a codon box, even if a base is substituted at the third codon position, because of degeneracy of the genetic code. Degeneracy can contribute to the formation of entirely new genes [25].

As an answer to the second question, I have further proposed the GNC code frozen-accident theory [1], suggesting that the primeval genetic code was established accidentally and frozen (Section 4.2).

However, the last problem still remains unsolved. The question is, “Why twenty natural amino acids are beautifully arranged in the universal genetic code table?”. In other words, “How were twenty amino acids arranged into the genetic code table?” (Figure 7). Two answers have been presented by other researchers as follows. 

Amino acids were arranged randomly or neutrally, using aaRSs and tRNAs, which were produced by the introduction of a small number of base substitutions into previously existing aaRS genes and tRNAs into the genetic code table after the establishment of the GNC primeval genetic code, as assumed by neutral theory [26].Amino acids, which were synthesized upon formation of a new metabolic pathway and accumulated in a cell at a high amount as assumed by the coevolution theory [23,24], were used and assigned into the genetic code table when the use of the amino acid was beneficial for cell growth as deduced by adaptive theory [27].

Answers to the last problem are described below.

1.An amino acid, which was newly synthesized through a new metabolic pathway and accumulated in a cell at a large amount, must be used for protein synthesis, as assumed by the coevolution theory [23,24] and was assigned as the adaptive theory expects [27]. That is supported by the fact that 2-ABA (α-amino-n-butylate), which has a simpler structure than Val and therefore should be synthesized with prebiotic means more easily than Val (Table 1), was not used in GNC code. Note that the reason why not only 2-ABA but also norvaline were not assigned into the code could not be reasonably explained.2.An amino acid newly synthesized through a new metabolic pathway should be assigned as a new amino acid into a previously existing genetic code table when productive power of primitive cells increased by use of the new amino acid. Inversely, if the amino acid inhibited cell growth, the amino acid should not take root as a new amino acid in the genetic code table.

Furthermore, the idea would be supported by the facts described below.

The reason why Glu was used after Asp is because Glu was synthesized with 2-oxoglutarate as a substrate, which was synthesized upon elongation of the metabolic pathway as a starting point of 2-oxyaloacetate, which is a substrate for Asp synthesis (Figure 8: step (1)).The use of Glu [E] induced the duplicated use of codons for three amino acids, Val, Ala, and Gly (Figure 8: step (2)), in order to suppress excess hydrophilicity of [GADVE]-protein upon the use of Glu.Three amino acids, Leu, Pro, and His, were selected and arranged as piling up onto the GNS code one by one after completion of the GNS code encoding five [GADVE]-amino acids (Figure 8: step (3)). The reason why a hydrophobic and α-helix forming Leu, a weakly hydrophobic and turn/coil forming Pro, and a weakly hydrophilic and α-helix forming His were used in a new genetic code table is because insufficient properties of GNS-encoding three amino acids, a hydrophobic and β-sheet forming Val, a weakly hydrophobic and α-helix forming Ala, and a hydrophilic and turn/coli forming Asp, could be complemented by capture of the three amino acids, Leu, Pro, and His.

Thus, it can be concluded that evolution of the genetic code from the GNC code to the universal genetic code proceeded as not the neutral theory, but as the coevolution theory [23,24] and the adaptive theory [27] expect.

## Figures and Tables

**Figure 1 genes-14-00375-f001:**
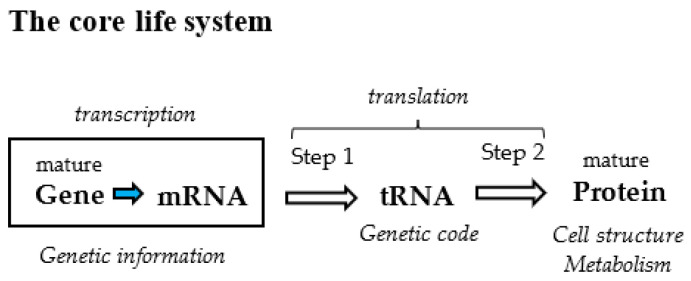
The core life system composed of four members, gene, tRNA (genetic code), and protein. tRNA and genetic code, which are underestimated in Central dogma, are brought to the surface in the core life system (Ikehara, 2021 [1]).

**Figure 2 genes-14-00375-f002:**
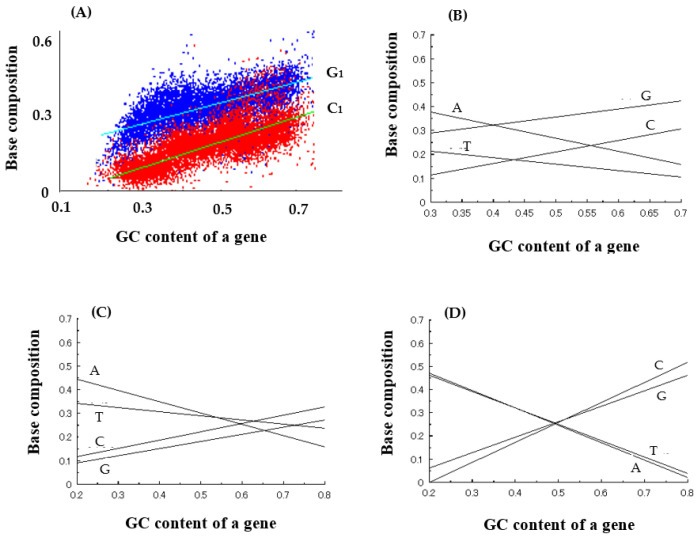
Average base composition of microbial genes. (**A**) Dependencies of G_1_ and C_1_ compositions at 1st codon position on GC content of a gene. (**B**) Dependencies of four base compositions (A, T, G, C) at 1st codon position on GC content of a gene. (**C**) Dependencies of four base compositions at 2nd codon position on GC content of a gene. (**D**) Dependencies of four base compositions at 3rd codon position on GC content of a gene. Note that the dependencies of base compositions on GC content are represented as straight lines approximated by the least-square method in (**B**–**D**).

**Figure 3 genes-14-00375-f003:**
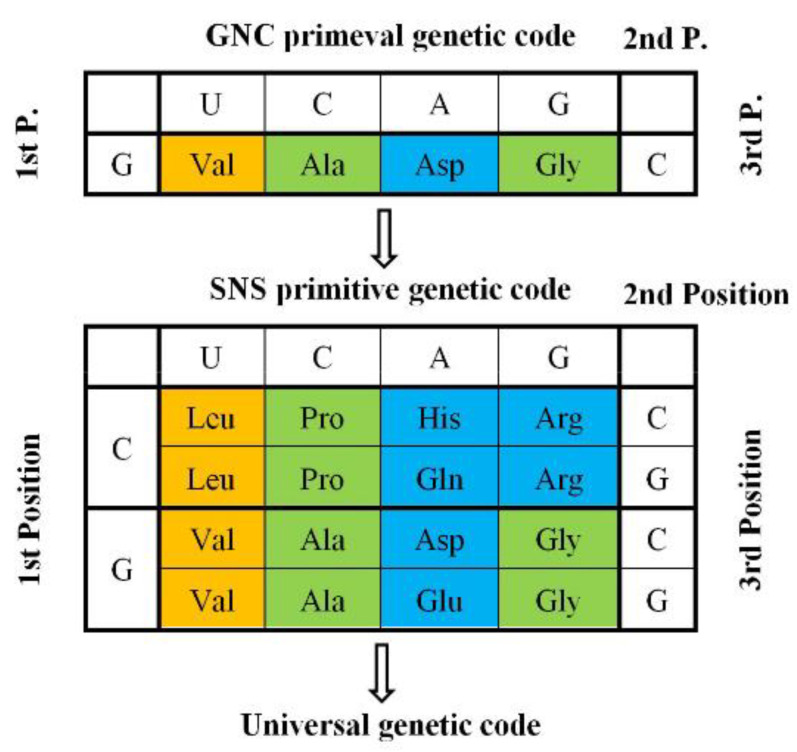
GNC-SNS primitive genetic code hypothesis [7]: The hypothesis, which was proposed based on formability of entirely new water-soluble globular protein, suggests that the universal genetic code originated from GNC primeval genetic code via SNS primitive genetic code. Amino acids written in brown, green, blue boxes indicate hydrophobic, intermediate, and hydrophilic amino acids, respectively.

**Figure 4 genes-14-00375-f004:**
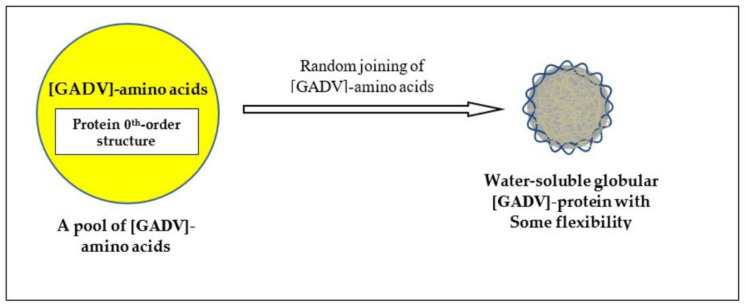
One of protein 0th-order structures or [GADV]-amino acid composition, which is the key concept of GADV hypothesis. Immature but water-soluble globular [GADV]-proteins, which were actually [GADV]-peptide aggregates, could be produced even by the joining of [GADV]-amino acids, which were randomly selected out from a pool containing [GADV]-amino acids at roughly equal amounts [16].

**Figure 5 genes-14-00375-f005:**
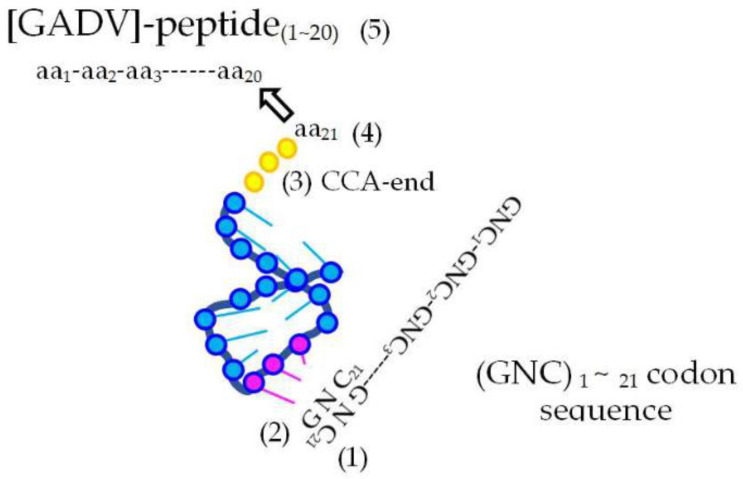
Elements involved in genetic code. (1) Codon on a (GNC)_n_ mRNA sequence. (2) Anticodon carried by primitive AntiC-SL tRNA. (3) CCA-end. (4) One of [GADV]-amino acids bound with 5-′CCA-3′ end of AntiC-SL tRNA. (5) Elongating oligo-[GADV]-peptide.

**Figure 6 genes-14-00375-f006:**
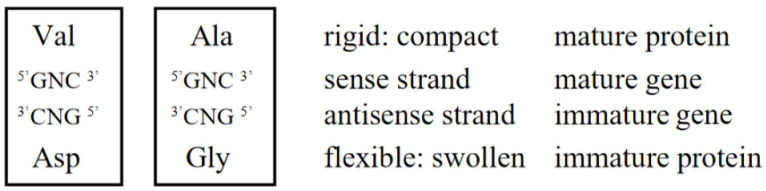
Four GNC codons were assigned to four [GADV]-amino acids frozen-accidentally. However, the assignment was not completely random. Two contrasting amino acid pairs (Val-Asp and Ala-Gly) were assigned to two complementary codon pairs (GUC-GAC and GCC-GGC) so that a rigid and compact mature [GADV]-protein could be formed from a flexible and swollen immature [GADV]-protein. That is, it is supposed that an immature [GADV]-protein composed of Asp and Gly at higher rates than Val and Ala evolved to a mature [GADV]-protein as accumulating amino acid replacements from Asp to Val and from Gly to Ala.

**Figure 7 genes-14-00375-f007:**
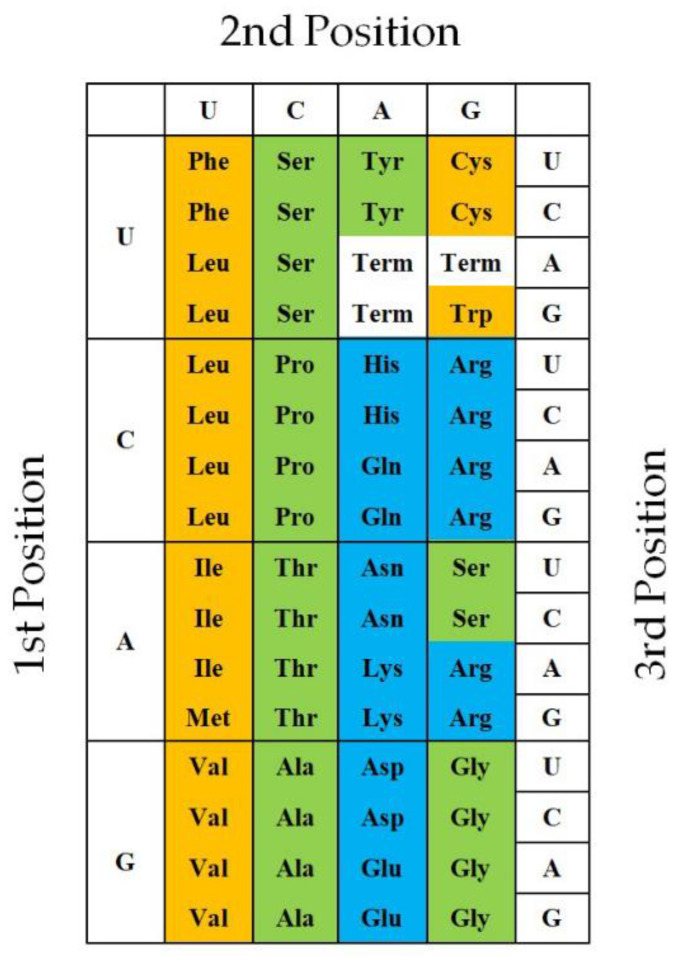
Twenty natural amino acids are characteristically arranged in the universal genetic code table at every three base positions of codon. Amino acids in brown, green and blue boxes mean hydrophobic, intermediate and hydrophilic amino acids.

**Figure 8 genes-14-00375-f008:**
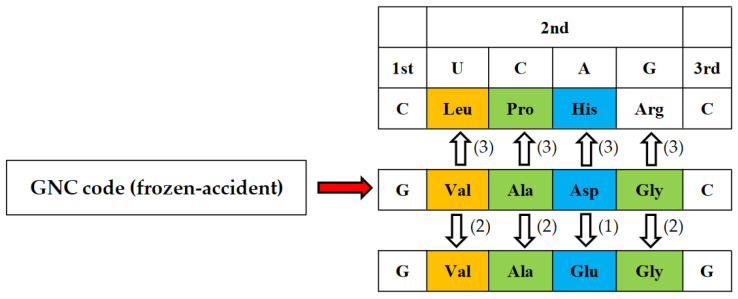
After GNC code was established as the GNC code frozen-accident theory assumes, the GNC code expanded from four GNC codons to capture GNG codons and CNC codons as coevolution theory [23,24] and adaptive theory [27] expects [1]. The order of the evolutionary pathway is indicated by numerals (1), (2), and (3). Amino acids in brown, green and blue boxes mean hydrophobic, intermediate and hydrophilic amino acids.

**Table 1 genes-14-00375-t001:** Experimental results obtained by Miller [10]. The results indicate that various natural and nonnatural amino acids should be produced with prebiotic means and accumulated on primitive Earth.

Compounds	Yield (mmol)	Compounds	Yield (mmol)
Gly	440	α-amino-n-butylate	270
Ala	790	α,γ-diaminobutylate	~30
Asp	34	β-amino-n-butylate	~0.3
Val	19.5	β-aminoisobutylate	~0.3
Glu	7.7	γ-aminobutylate	2.4
Leu	11.3	Norvaline	61
Ile	4.8	Norleucine	6
Pro	1.5	β-alanine	18.8
Ser	5	Alloisoleucine	5.1
Thr	~0.8	Isoserine	5.5

**Table 3 genes-14-00375-t003:** Base sequences of *E. coli* anticodon stem loops (AntiC-SLs) decoding GNC codons to [GADV]-amino acids. Bases chemically modified are shown with green or brown bold letters or green symbols as “;” and “/”, which are quoted from the tRNA database [19]. It is supposed that those chemically modified bases around anticodon play roles in protruding anticodon from AntiC-loop and in assisting of binding of the anticodon (written in red bold letters) with the codon. Base sequences of five AntiC-SL RNAs, which are used for translation to GNS-encoding five amino acids are given in upper table. Three amino acids, Gly, Ala, and Val, having no chemically modified base, are written in red bold letters Base sequences of five AntiC-SL RNAs, which are used for translation of five C-start codons-encoding five amino acids, Leu, Pro, His, Gln, and Arg, are given in lower table.

	**5′-AntiC-Stem**	**AntiC-Loop**	**3′-AntiC-Stem**
Gly	CGACC	UUGCCAA	GGUCG
Ala	CUUGC	AUGGCAU	GCAAG
Asp	CCUGC	CUQUC/C	GCAGG
Val	CCACC	UUGACAU	GGUGG
Glu	CCGCC	AUGGCAU	GGCGG
	**5′-AntiC-Stem**	**AntiC-Loop**	**3′-AntiC-Stem**
Leu	CUAGC	UUCAG;P	GPUAG
Pro	CUUCG	JUCGGKA	CGAAG
His	CUGGA	UUQUG/P	PCCAG
Gln	CCGGA	JUCUG/P	PCCGG
Arg	CUCGG	UUCAG;P	GPUAG

**Table 4 genes-14-00375-t004:** Stability of complementary triplets (codon/anticodon). Codons encoding [GADV]-amino acids strongly bind with the respective anticodons except Ser (AGC/GCT). This suggests that stable base pairs could be formed between codon-anticodon even before a chemical modification system had been established. These data are quoted from Taghavi et al. [20].

Amino Acid	Codon/Anticodon	ΔG_t_/k_B_T
**Gly [G]**	GGC/GCC	1.71
**Ala [A]**	GCC/GGC	1.71
Ser	AGC/GCT	1.86
**Val [V]**	GTC/GAC	2.01
**Asp [D]**	GAC/GTC	2.02
Thr	ACC/GGT	2.07
Arg	AGA/TCT	2.18
Phe	TTC/GGT	2.2
Glu	GAA/TTC	2.2
Asn	AAC/GTT	2.25

## Data Availability

No new data were created in this review article.

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
