# Peer review of "Why Were [GADV]-amino Acids and GNC Codons Selected and How Was GNC Primeval Genetic Code Established?"

_genes, 2023, doi:10.3390/genes14020375_

Round 1
Reviewer 1 Report
This is an interesting and well-written manuscript.
To improve it I suggest the following:
- The author has to discuss that the GNC code is a code predict by the coevolution theory.
Author Response
My responses to the reviewer 1
Comments and Suggestions for Authors
This is an interesting and well-written manuscript.
To improve it I suggest the following:
- The author has to discuss that the GNC code is a code predict by the coevolution theory.
Thank you very much for your valuable comments. One sentence, “In addition, both amino acids, which are encoded by GNC code, and amino acids, which were synthesized in the earliest phase of amino acid metabolism, were [GADV]-amino acids [1]. Therefore, the idea, that amino acids were used in the first genetic code, GNC, is consistent with the coevolution theory [23,24].”, was added from line 545 to 548 in Discussion according to your comment.

Reviewer 2 Report
This paper by Kenji Ikehara describes possible explanations for the selection of the so-called GADV amino acids and GNC codons through the primitive genetic code. This question is certainly of significant interest: the coevolution of nucleotides and amino acids at some point must have given rise to the genetic code (i.e. codon sequence), but the circumstances surrounding this process are unclear. Previous work by the author has been significant in elucidating the GADV and GNC hypothesis.
1) At times, the text of the paper can be a little informal, which detracts from the science merit of the work, in my opinion. I recognize that the paper is written by a sole author, thus use of “I” may be hard to eliminate completely. But my understanding is that “I” in scientific writing is generally frowned upon. In addition to the use of I, some text comes across informally, such as on page 2 (“At the beginning, I started from the study on origin of modern genes, independently…”), page 10 (“During the above considerations, an idea occurred to me.”), page 12 (“My answers to the last problem are as follows.”), page 12 (“Furthermore, the idea of mine would be supported….”). I understand that the author pioneered much of the previous work on this hypothesis, but writing more formally may help the reader follow precisely what this paper is adding to the theory, rather than what has been described previously in the literature.
2) The paper uses the terminology “primeval” and “primitive” for various genetic codes. It was unclear if these words are referring to the same genetic code and the same point in time, or if they are distinct. Clarifying the language regarding the genetic code and its evolution would aid the reader’s understanding.
3) What is being argued vs. what is established and agreed on in the community is somewhat hard to follow. Including more references (if possible) and explicitly stating when mentioning a hypothesis from the author’s previous work or this paper would be beneficial.
4) Please see attached document for minor wording changes or typos and some additional notes/questions.

Author Response
My responses to the reviewer 2
Thank you very much for your kind and appropriate reviewing. I greatly appreciate your valuable comments and suggestions. The manuscript was revised according to all your comments. All modifications were highlighted in red in the revised manuscript. Furthermore, I reply to your comments point-by-point as described below.
Comments and Suggestions for Authors
This paper by Kenji Ikehara describes possible explanations for the selection of the so-called GADV amino acids and GNC codons through the primitive genetic code. This question is certainly of significant interest: the coevolution of nucleotides and amino acids at some point must have given rise to the genetic code (i.e. codon sequence), but the circumstances surrounding this process are unclear. Previous work by the author has been significant in elucidating the GADV and GNC hypothesis.
1) At times, the text of the paper can be a little informal, which detracts from the science merit of the work, in my opinion. I recognize that the paper is written by a sole author, thus use of “I” may be hard to eliminate completely. But my understanding is that “I” in scientific writing is generally frowned upon. In addition to the use of I, some text comes across informally, such as on page 2 (“At the beginning, I started from the study on origin of modern genes, independently…”), page 10 (“During the above considerations, an idea occurred to me.”), page 12 (“My answers to the last problem are as follows.”), page 12 (“Furthermore, the idea of mine would be supported….”). I understand that the author pioneered much of the previous work on this hypothesis, but writing more formally may help the reader follow precisely what this paper is adding to the theory, rather than what has been described previously in the literature.
Several sentences were changed as described below, in order to avoid the use of the words, “I”, “My” and “Mine” according to the comments of Reviewer 2.
- The sentence, “At the beginning, I started from the study on origin of modern genes, independently of origins of the genetic code and life, if entirely new genes are actually generated still now.”, was changed to “The study on the GADV hypothesis started from the origin of modern genes independently of the origins of the genetic code and life.” (see from Line 74 to 75).
- The sentences, “During the above considerations, an idea occurred to me. The idea is GNC code freeze-accident theory, suggesting that only the correspondence relations between GNC codons and [GADV]-amino acids were accidentally determined and frozen after the correspondence relations were established [1].”, were changed to “During the above considerations, the GNC code freeze-accident theory was conceived, suggesting that only the correspondence relations between GNC codons and [GADV]-amino acids were accidentally determined and frozen after the correspondence relations were established [1]. Establishment process of the primeval genetic code could be consistently explained by incorporating the GNC code freeze-accident theory [1].”
- The sentence, “My answers to the last problem are as follows.”, was changed to “Answers to the last problem are described below.” (see Line 597).
- The phrase, “of mine”, was deleted from the sentence, “Furthermore, the idea of mine would be supported by the facts described below.”, as “Furthermore, the idea would be supported by the facts described below.” (see Line 633).
- Furthermore, I added the sentence below to indicate more clearly what this paper is adding to the theory (see from Line 170 to 173). “In this article, establishment process of the GNC primeval genetic code is discussed with a focus on the latter aspect, “How were amino acids and codons selected for the first genetic code and how was the first genetic code established?”, as summarizing the results obtained so far.”.
2) The paper uses the terminology “primeval” and “primitive” for various genetic codes. It was unclear if these words are referring to the same genetic code and the same point in time, or if they are distinct. Clarifying the language regarding the genetic code and its evolution would aid the reader’s understanding.
My answer is as follows.
In the manuscript, the adjective, “primeval”, is used to show that the GNC code was established at earlier time than the SNS code. Accordingly, the terms, SNS primitive genetic code and GNC-SNS primitive genetic code, were used in the manuscript.
3) What is being argued vs. what is established and agreed on in the community is somewhat hard to follow. Including more references (if possible) and explicitly stating when mentioning a hypothesis from the author’s previous work or this paper would be beneficial.
The sentence “In this article, establishment process of the GNC primeval genetic code is discussed with a focus on the latter aspect, “How were amino acids and codons selected for the first genetic code and how was the first genetic code established?”, as summarizing the results obtained so far.” was added to answer to the question of Reviewer 2.
4) Please see attached document for minor wording changes or typos and some additional notes/questions.
Thank you very much again. I sincerely appreciate your wording changes or typos and some additional notes/questions. The manuscript was revised according to all your points indicated except three points, about “dot-plot”, GC-NSF(a) and “to that”.
1. Line 45: The meaning “GC-NSF(a)” is explained from two lines below.
- Line 312-314: The sentence, "It was confirmed be dot-plot analysis that imaginary [GADV]-proteins do satisfy the four conditions for water-soluble globular protein synthesis [1,7,15].", was changed to "It was confirmed by analysis, in which points are given when imaginary [GADV]-proteins satisfy the four conditions for water-soluble globular protein synthesis [1,7,15].".
- Line 505: The phrase, "to that", was changed to "to the thing that".

Reviewer 3 Report
The manuscript does not have enough coherence in writing. There are many repetitive concepts.
Author Response
Comments for Author
The manuscript does not have enough coherence in writing.
I extensively revised the manuscript to have enough coherence (please see the attachment PDF file).
There are many repetitive concepts.
To answer to the indication of Reviewer 3, the sentence “In this article, establishment process of the GNC primeval genetic code is discussed with a focus on the latter aspect, “How were amino acids and codons selected for the first genetic code and how was the first genetic code established?”, as summarizing the results obtained so far.” was added (please see from Line 170 to 173).

Round 2
Reviewer 2 Report
The author has done an adequate job responding to my comments.